# Association between gender social norms and cardiovascular disease mortality and life expectancy: an ecological study

Iona Lyell ,[1] Sadiya S Khan,[2] Mark Limmer,[1] Martin O'Flaherty,[3] Anna Head[3]

[1]Faculty of Health and Medicine, Lancaster University, Lancaster, UK
[2]Feinberg School of Medicine, Northwestern University, Chicago, Illinois, USA
[3]Department of Public Health, Policy and Systems, University of Liverpool, Liverpool, UK

**Correspondence to**
Anna Head;
Anna.Head@liverpool.ac.uk

## ABSTRACT

**Objective** Examine the association between country-level gender social norms and (1) cardiovascular disease mortality rates; (2) female to male cardiovascular disease mortality ratios; and (3) life expectancy.

**Design** Ecological study with the country as the unit of analysis.

**Setting** Global, country-level data.

**Participants** Global population of countries with data available on gender social norms as measured by the Gender Social Norms Index (developed by the United Nations Development Programme).

**Main outcome measures** Country-level female and male age-standardised cardiovascular disease mortality rates, population age-standardised cardiovascular disease mortality rates, female to male cardiovascular disease mortality ratios, female and male life expectancy at birth. Outcome measure data were retrieved from the WHO and the Institute for Health Metrics and Evaluation. Multivariable linear regression models were fitted to explore the relationship between gender social norms and the outcome variables.

**Results** Higher levels of biased gender social norms, as measured by the Gender Social Norms Index, were associated with higher female, male and population cardiovascular disease mortality rates in the multivariable models (β 4.86, 95% CIs 3.18 to 6.54; β 5.28, 95% CIs 3.42 to 7.15; β 4.89, 95% CIs 3.18 to 6.60), and lower female and male life expectancy (β −0.07, 95% CIs −0.11 to −0.03; β −0.05, 95% CIs −0.10 to −0.01). These results included adjustment within the models for potentially confounding country-level factors including gross domestic product per capita, population mean years of schooling, physicians per 1000 population, year of Gender Social Norms Index data collection and maternal mortality ratio.

**Conclusions** Our analysis suggests that higher levels of biased gender social norms are associated with higher rates of population cardiovascular disease mortality and lower life expectancy for both sexes. Future research should explore this relationship further, to define its causal role and promote public health action.

## INTRODUCTION

The population health benefits of gender equity are well recognised within the global

### STRENGTHS AND LIMITATIONS OF THIS STUDY

⇒ The Gender Social Norms Index has been used as a quantified measure of gender social norms at country level for 75 countries.
⇒ The outcome measures of cardiovascular disease mortality and life expectancy were stratified by men and women.
⇒ Confounding variables were included within the analysis to take into account underlying differences between countries.
⇒ Data was not available to analyse the relationship between variables over time.

health research community and reflected in development priorities including the Sustainable Development Goals.[1 2] However, research in the area of gender social norms and health has thus far been largely focused on issues classically considered as 'women's health' related (reproduction, child health and intimate partner violence[3–14]) and there is a lack of understanding in terms of the relationship with non-communicable diseases at a population level. It is cardiovascular disease (CVD) specifically that is the leading cause of death worldwide[15] and a significant cause of premature mortality globally for both women and men.[15]

Gender and the norms associated with it are considered a construct of society, influenced by but separate from biological sex[16] (see box 1). A means through which power dynamics flow within societies, gender determines actions and reactions of individuals and communities and therefore acts as a social determinant of health.[16 17] Gender social norms are the 'unspoken rules' influencing how individuals enact their gender and interact with others.[6 18] Theoretical explanations exist to link gender and gender social norms to CVD outcomes including in healthcare, health behaviours and wider determinants of health.[19] These include a

**Box 1   Glossary box**

Sex—biologically determined characteristics (genetics, hormones, anatomy) that lead to categorisation of individuals, typically into male, female and intersex.

Gender—considered a social construct, sociologically determined understandings and behaviours interacting with but separate from sex. Influenced by societal norms and changes over time.

Social norms—the unwritten societal expectations influencing how we think and behave in a social situation, tend to be considered on an individual level, that is, how individuals react to what societal expectations exist.

Gender norms—engrained understandings of how individuals should act based on their gender, learnt through institutions such as family, school, workplace, religion, media and other social institutions. Conceptualised as a force external to individuals, upheld by institutions. Contributes to the perpetuation of the gender system which on the whole tends to favour men and masculinity.

Gender social norms—a combining of the social norms and gender norms concepts. Includes the individual and societal beliefs about acceptable behaviour based on gender and how that is borne out in society. These norms are embedded and upheld by institutions within society.

recognition of gender bias in research and treatment of CVD, an understanding that gender social norms influence patterns of CVD risk factors, for example, stress and smoking, and their impact on important social determinants of CVD including education and employment.[17 20 21] Therefore, while there are theoretical explanations for a connection between gender social norms and CVD mortality at a population level,[19] this has not yet been explored in an ecological framework with use of country-level data.

It is theorised that increased levels of biased gender social norms may be associated with determinants and behaviours harmful to cardiovascular health on a population-level, and therefore an increased CVD burden and reduced life expectancy more broadly for both women and men.[19 22] Therefore, this study aimed to explore the association between gender social norms, as measured by the Gender Social Norms Index and CVD mortality and life expectancy at the country level.

## METHODS

This research made use of an ecological study design with the country as the unit of analysis to investigate the relationship between gender social norms and CVD mortality rates, female to male CVD mortality ratios and life expectancy.

### Outcome variables

The outcome variables were defined as; female, male and total age-standardised CVD mortality rate per 100000 population, female to male CVD mortality ratio, female and male life expectancy at birth in years. The CVD outcome data was derived from the Global Burden of Disease (GBD) study 2019.[15] Age-standardised rates were calculated with use of the GBD world population age standard,[15] calculated from the GBD 2019 population estimates.[23] CVD as cause of death in the GBD 2019 includes 11 disease categories: ischaemic heart disease, stroke, hypertensive heart disease, atrial fibrillation/flutter, rheumatic heart disease, non-rheumatic valvular heart disease, cardiomyopathy and myocarditis, aortic aneurysm, peripheral artery disease, endocarditis and other cardiovascular and circulatory diseases.[15]

Life expectancy at birth data was downloaded from the Global Health Observatory hosted by the WHO.[24] It is defined as 'the average number of years that a newborn could expect to live, if he or she were to pass through life exposed to the sex and age specific death rates prevailing at the time of his or her birth, for a specific year, in a given country, territory, or geographic area'.[24] Three countries (Andorra, UK and Palestine) did not have data available from this source, instead this data was retrieved from the Institute for Health Metrics and Evaluation.[25]

### Primary explanatory variable

The Gender Social Norms Index (GSNI) was created by the United Nations Development Programme and is available to download from http://hdr.undp.org/en/gsni. It is a novel, freely available, quantitative measure of gender social norms at the country level and derived from the World Values Survey.[26] The data was collected in two waves of the survey undertaken between 2005–2009 and 2010–2014. The GSNI is made up of four domains (political, educational, economic and physical integrity). The share of people with at least two gender biases (GSNI2) was chosen as the primary explanatory variable as it was presumed this would demonstrate a stronger relationship if there was one to be found compared with the share of people with one bias. Values are provided as a percentage for each country. The value for GSNI2 can be interpreted as an estimate of the prevalence in a country of moderate-to-intense gender bias.[27] Seventy-five countries have an index value available which amounts to the representation of 81% of the global population.[27]

### Covariates

Gross domestic product (GDP) per capita, physicians per 1000 population and mean years of schooling were included as additional country-level covariates based on evidence of the association of these variables with population health.[28–30] Data sets for these variables were retrieved from The World Bank and Our World in Data.[31–33] Maternal mortality ratio was included as a covariate in the female life expectancy model and was retrieved from The World Bank.[34] It is possible that the wave of GSNI data collection may have influenced the results, for example, through passage of time or changes in methodology. Therefore, this variable was also included within the model.

## Choice of variables (year)

Variables (both outcome and explanatory) collected for the year 2014 or closest approximation were selected as the GSNI collection period ranged from 2005 to 2014. This included CVD mortality, GDP per capita, physicians per 1000 population, mean years of schooling and maternal mortality ratio. Life expectancy data from 2015 was used due to availability. If a value was not available for the year of analysis for one of the explanatory variables, a value for the closest year available was inserted. If years either side of the missing year were available, the most contemporaneous year was selected.

## Statistical analysis

Normally distributed continuous variables were summarised using mean and SD. If not normally distributed, variables were summarised using median and IQRs. Categorical variables were summarised using frequency tables. The GSNI values were compared over time for countries with values for both collection waves, and the values compared between collection waves. Differences in these groups were assessed visually with the use of box plots and statistically using appropriate parametric or non-parametric hypothesis tests after assessing the normality and variance of the data.

Scatter plots were used to visualise the relationships between each of the outcome and explanatory variables. If the relationship was not linear, consideration of variable transformation was made for further analysis.

Univariable linear regression models were computed for each of the three outcome variables and each of the explanatory variables in turn. Multivariable linear regression models were then created for each outcome variable using the ordinary least squares method.[35] Testing of the null hypothesis ($\beta_1=0$) was done with use of the t-distribution where $t=\beta/SE$ and SE is the SE of the slope. Model goodness-of-fit was assessed with use of $R^2$ and the F statistic.[35 36]

To act as a sensitivity analysis to the choice of data time points used, the analysis was repeated for outcome and covariate data collected for 2017 (2019 for life expectancy) as this is the most recent year for which the majority of data points were available. Sensitivity analysis was also conducted with the use of GSNI1 values to test the choice of primary explanatory variable.

The assumptions for linear regression models (linearity, collinearity, homoscedasticity and normality of errors) were assessed with use of appropriate plots. Residuals versus leverage plots and Cook's distance were used to identify potential outliers or influential observations.

The significance level for p values was set at 0.05 for all statistical tests. The statistical assumptions required for linear regression models were tested and deemed acceptable for all models. All data management and analysis were conducted using R V.4.1.2.

## Patient and public involvement

No members of the public or patients were involved in the design or implementation of the study.

## RESULTS

Summary statistics for the 75 countries with GSNI values available can be viewed in table 1. Female age-standardised CVD mortality rate ranged from 62 per 100 000 to 1025 per 100 000 (IQR 128–345). Male age-standardised CVD mortality rate ranged from 107 per 100 000 to 1205 per 100 000 (IQR 180–429). Age-standardised CVD mortality for the total population ranged from 83 to 1107 per 100 000 (IQR 156–373). Japan had the lowest male, female and total CVD mortality rates and Uzbekistan the highest. The ratio of female to male CVD mortality rates ranged from 0.5 in Vietnam to 1.6 in Qatar (IQR 0.67–0.85). Female life expectancy ranged from 61 to 86.4 years (IQR 74.5–82.6). Zimbabwe had the lowest female life expectancy and Japan the highest. Male life expectancy ranged from 55.7 years in Zimbabwe to 80.7 years in both Japan and Singapore (IQR 67.9–77.6).

The GSNI value for 2 or more biases ranged from 7.4% in Andorra to 98.1% in Pakistan (median 68.6, IQR 41.9–86.3). Thirty-one countries had a GSNI2 value available for both collection periods, with no statistically significant difference between the values across the two time periods (paired t-test=0.104, p value=0.92) (online supplemental figure S1). Comparison was made between the GSNI2 values included in the analysis (n=75) collected for each time period (online supplemental figure S1). Those collected during the later period were statistically significantly higher than those collected earlier (Mann-Whitney U test=320, p value=0.02).

GSNI2 was positively correlated with female CVD mortality rates (rho=0.67, p value<0.05), with male CVD mortality rates (rho 0.53, p value<0.05) and with population CVD mortality rates (rho=0.60, p value<0.05). GSNI2 was also positively correlated with female to male CVD mortality ratio (rho=0.57, p value<0.05) (figure 1). GSNI2 was negatively correlated with female life expectancy (rho=−0.72, p value<0.05) and with male life expectancy (rho=−0.61, p value≤0.05) (figure 2). Please see online supplemental table S1 and figures S2,3 for the list of countries included and additional scatter plot versions with points labelled by country.

Seventy-five countries had data available for the univariable regression analyses. Sixty-seven Countries had data available across all covariates for the multivariable regression analyses. Andorra was not included in the female life expectancy outcome as maternal mortality ratio not available (n=66); Qatar was excluded from the CVD ratio outcome analysis because it was identified as an influential outlier (n=66). The list of included countries within each analysis can be found in online supplemental tables 1 and 2.

In the univariable models, higher levels of gender bias as measured by the GSNI2 were statistically significantly

**Table 1** Summary statistics

| Summary statistics | n=75 |
| --- | --- |
| Female CVD age adjusted mortality rate (per 100 000 population) Year: 2014 Source: Global Burden of Disease 2019 | |
| Min | 62 |
| Max | 1024.9 |
| Median (IQR) | 241.30 (128.06–345.30) |
| Missing | 0 |
| Male CVD age adjusted mortality rate (per 100 000 population) Year: 2014 Source: Global Burden of Disease 2019 | |
| Min | 106.8 |
| Max | 1205.2 |
| Median (IQR) | 278.05 (180.32–428.51) |
| Missing | 0 |
| Population CVD age adjusted mortality rate (per 100 000 population) Year: 2014 Source: Global Burden of Disease 2019 | |
| Min | 82.6 |
| Max | 1107 |
| Median (IQR) | 262.98 (155.95–372.59) |
| Missing | 0 |
| Ratio of female to male CVD mortality rates Year: 2014 Source: Global Burden of Disease 2019 | |
| Min | 0.5 |
| Max | 1.6 |
| Median (IQR) | 0.72 (0.67–0.85) |
| Missing | 0 |
| Female life expectancy at birth Year: 2015 Source: World Health Observatory | |
| Min | 61 |
| Max | 86.4 |
| Median (IQR) | 78.60 (74.45–82.60) |
| Missing | 0 |
| Male life expectancy at birth Year: 2015 Source: World Health Observatory | |
| Min | 55.7 |
| Max | 80.7 |
| Median (IQR) | 73.00 (67.90–77.60) |
| Missing | 0 |
| GSNI index two or more bias Year: 2005–2014 Source: Gender Social Norms Index | |
| Min | 7.4 |

Continued

**Table 1** Continued

| Summary statistics | n=75 |
| --- | --- |
| Max | 98.1 |
| Median (IQR) | 68.56 (41.94–86.33) |
| Missing | 0 |
| Year of index collection Source: Gender Social Norms Index | |
| 2005–2009 | 18 (24) |
| 2010–2014 | 57 (76) |
| Physicians per 1000 population Year: 2014 Source: World Bank | |
| Min | 0 |
| Max | 5 |
| Median (IQR) | 71; 2.30 (1.29–3.22) |
| Missing | 4 |
| GDP per capita Year: 2014 Source: World Bank | |
| Min | 566.9 |
| Max | 97 019.2 |
| Median (IQR) | 68; 9181 (4094–31 623) |
| Missing | 7 |
| Maternal mortality ratio Year: 2014 Source: World Bank | |
| Min | 2 |
| Max | 943 |
| Median (IQR) | 70; 25.50 (10.25–82.75) |
| Missing | 5 |
| Mean years of schooling Year: 2014 Source: Our World in Data | |
| Min | 1.4 |
| Max | 14 |
| Median (IQR) | 69; 10.10 (7.60–11.80) |
| Missing | 6 |

CVD, cardiovascular disease; GDP, gross domestic product; GSNI, Gender Social Norms Index.

associated with higher female CVD mortality rates (β 3.63, 95% CIs 2.48 to 4.78), male CVD mortality rates (β 3.57, 95% CIs 2.03 to 5.12) and population CVD mortality rates (β 3.57, 95% CIs 2.29 to 4.85). The statistically significant association remained after adjusting for physicians per 1000 population, mean years of schooling, GDP per capita and GSNI data collection period (tables 2 and 3). The multivariable model results indicate that for every 1% greater proportion of the population who hold two or more gender biasses, the female CVD mortality rate was higher by 4.86 per 100 000 (95% CI 3.18 to 6.54),

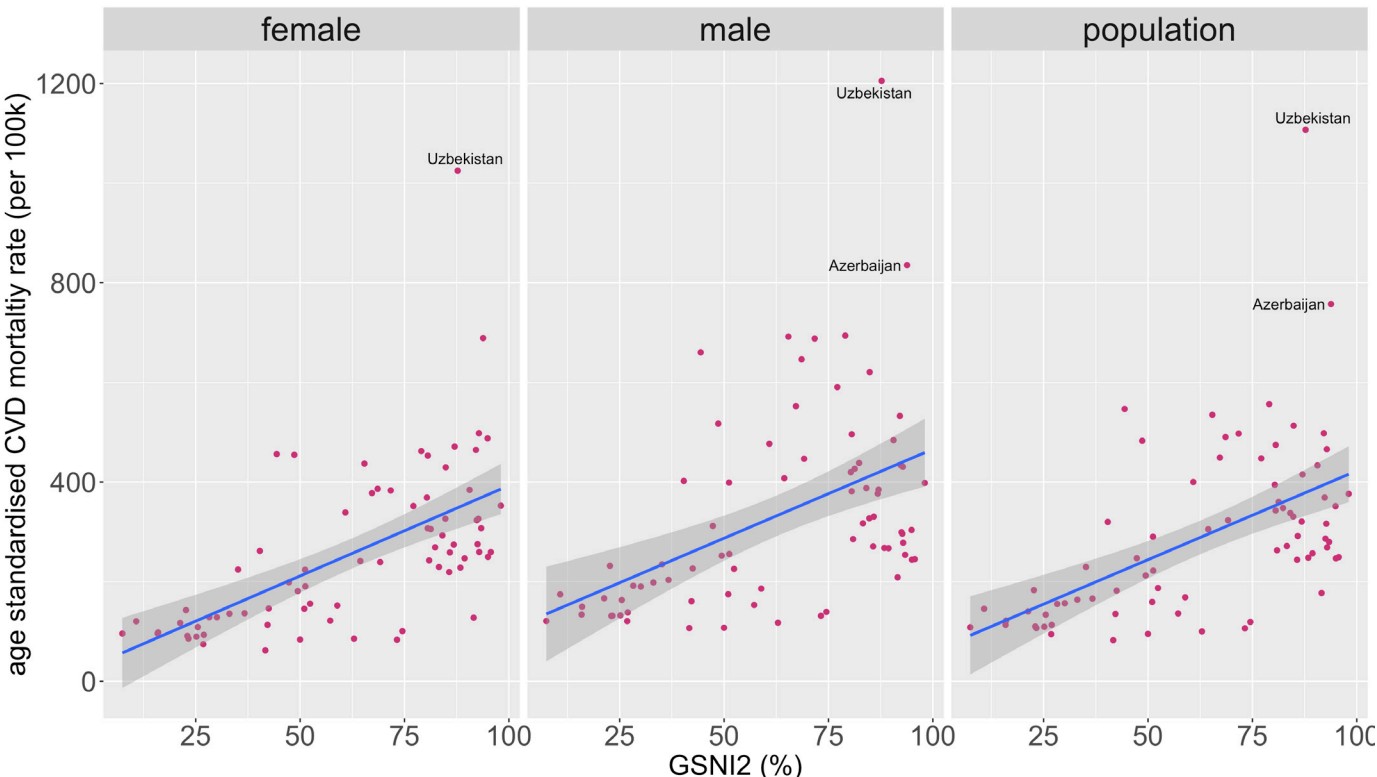

**Figure 1** Scatter plots of GSNI2 against female, male and population age-standardised CVD mortality rates. CVD, cardiovascular disease; GSNI2, Gender Social Norms Index 2.

male CVD mortality rate higher by 5.28 per 100 000 (95% CI 3.42 to 7.15) and the population CVD mortality rate was higher by 4.89 per 100 000 (95% CI 3.18 to 6.60). In

the univariable model GSNI2 was statistically significantly associated with female to male CVD mortality ratio (β 0.003, 95% CIs 0.002 to 0.005). This relationship was not

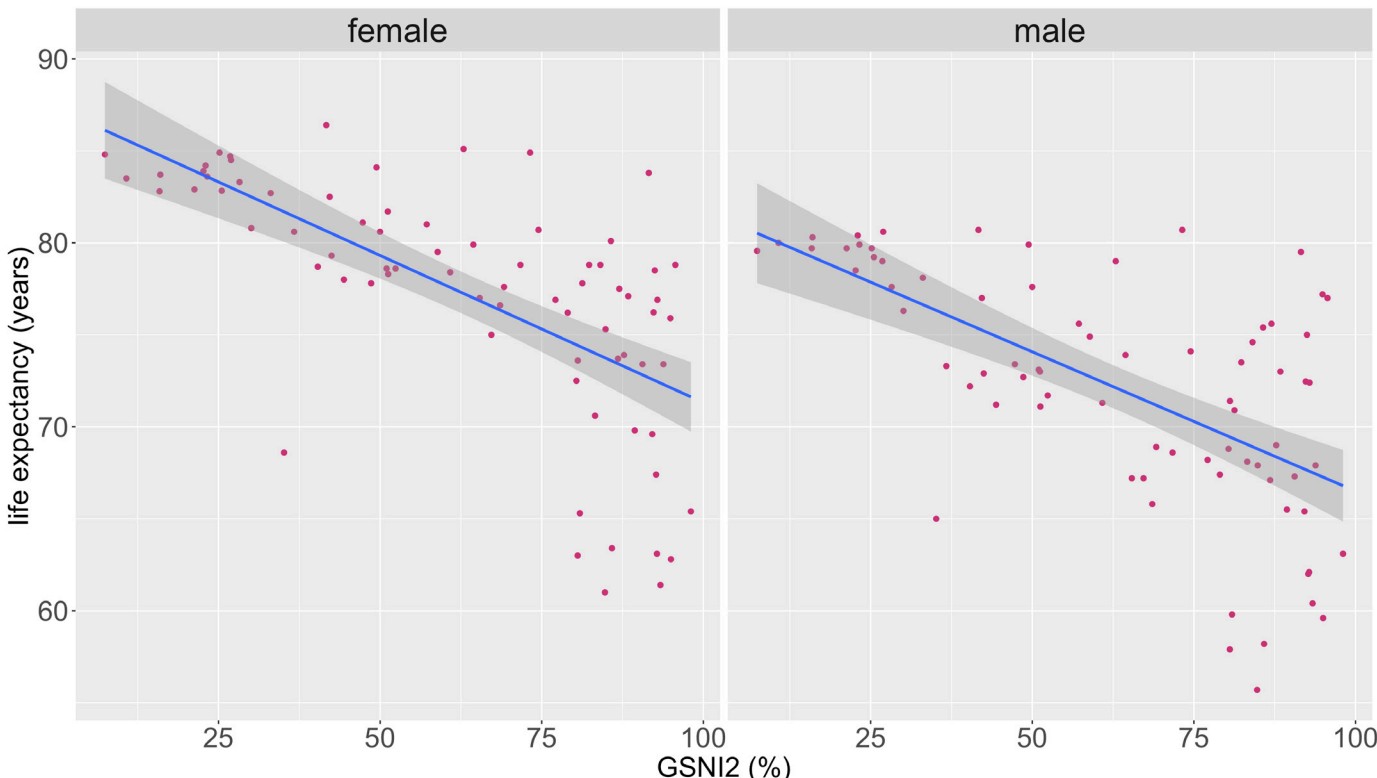

**Figure 2** Scatter plots of GSNI2 against female and male life expectancy. GSNI2, Gender Social Norms Index 2.

**Table 2** Results of univariable and multivariable regression models for the female and male CVD mortality outcomes

| | Dependent variable | | | |
| --- | --- | --- | --- | --- |
| | Female CVD mortality 2014 | | Male CVD mortality 2014 | |
| | (1) | (2) | (3) | (4) |
| Constant | 30.04 (39.25) | −133.15 (94.46) | 108.45* (52.87) | −184.13 (104.74) |
| GSNI2 | 3.63*** (0.58) | 4.86*** (0.84) | 3.57*** (0.78) | 5.28*** (0.93) |
| Physicians per 1000 | | 32.31 (19.26) | | 70.59** (21.35) |
| Mean years of schooling | | 13.75 (9.37) | | 21.71* (10.39) |
| GDP per capita | | −0.002* (0.001) | | −0.004*** (0.001) |
| GSNI data collection period 2010–2014 | | −97.89* (40.74) | | −127.18** (45.17) |
| $R^2$ | 0.35 | 0.49 | 0.23 | 0.56 |
| Adjusted $R^2$ | 0.34 | 0.44 | 0.21 | 0.53 |
| F statistic | 39.68*** (df=1; 73) | 11.54*** (df=5; 61) | 21.23*** (df=1; 73) | 15.62*** (df=5; 61) |

*P values<0.05, **p<0.01,***p<0.001.
Numbers in brackets are SEs.
CVD, cardiovascular disease; GDP, gross domestic product; GSNI2, Gender Social Norms Index 2.

significant within the multivariable model (β 0.001, 95% CIs −0.001 to 0.003).

Table 4 demonstrates the model results for the life expectancy outcomes. In the univariable analysis, higher levels of gender bias as measured by the GSNI2 were statistically significantly associated with lower female life expectancy (β −0.16, 95% CIs −0.20 to −0.12) and lower male life expectancy (β −0.15, 95% CIs −0.20 to −0.11). This relationship remained statistically significant after controlling for physicians per 1000 population, mean years of schooling, GDP per capita and GSNI data collection period. Maternal mortality ratio was also included within the female life expectancy model. Within the life expectancy multivariable models, the physicians per 1000 and GDP per capita variables were log transformed due

to their relationship with the outcome variable. In the final life expectancy model the coefficient indicates that for every 1% increase in the GSNI2, female life expectancy was lower by 0.07 years (25.6 days) (95% CI −0.11 to −0.03) and male life expectancy was lower by 0.05 years (18.3 days) (95% CI −0.10 to −0.01).

The sensitivity analysis made use of data from 2017 to 2019 with no change in direction of association between each outcome variable and GSNI2. The only change in statistical significance compared with the primary analysis was the multivariable male life expectancy model in which male life expectancy was not found to be statistically significantly associated with GSNI2 (online supplemental tables S3–S5). GSNI1 and GSNI2 were strongly positively correlated (rho=0.98, p value≤0.05) and the

**Table 3** Results of univariable and multivariable regression models for CVD mortality and female to male CVD mortality ratio

| | Dependent variable | | | |
| --- | --- | --- | --- | --- |
| | CVD mortality 2014 | | Female to male CVD mortality ratio 2014 | |
| | (1) | (2) | (3) | (4) |
| Constant | 65.58 (43.71) | −140.69 (95.92) | 0.57*** (0.05) | 0.90*** (0.09) |
| GSNI2 | 3.57*** (0.64) | 4.89*** (0.85) | 0.003*** (0.001) | 0.001 (0.001) |
| Physicians per 1000 | | 47.45* (19.55) | | −0.04* (0.02) |
| Mean years of schooling | | 17.25 (9.52) | | −0.01 (0.01) |
| GDP per capita | | −0.003** (0.001) | | −0.0000 (0.0000) |
| GSNI data collection period 2010–2014 | | −110.79** (41.36) | | −0.03 (0.04) |
| $R^2$ | 0.30 | 0.53 | 0.24 | 0.46 |
| Adjusted $R^2$ | 0.29 | 0.49 | 0.22 | 0.42 |
| F statistic | 30.97*** (df=1; 73) | 13.54*** (df=5; 61) | 22.44*** (df=1; 73) | 10.30*** (df=5; 60) |

Numbers in brackets are SEs.
*P values<0.05, **p<0.01,***p<0.001.
CVD, cardiovascular disease; GDP, gross domestic product; GSNI2, Gender Social Norms Index 2.

**Table 4** Results of univariable and multivariable regression models for female life expectancy at birth and male life expectancy at birth

| | Dependent variable | | | |
| | Female life expectancy 2015 | | Male life expectancy 2015 | |
| | (1) | (2) | (3) | (4) |
|---|---|---|---|---|
| Constant | 87.31*** (1.46) | 68.74*** (4.82) | 81.65*** (1.51) | 50.48*** (5.98) |
| GSNI | −0.16*** (0.02) | −0.07*** (0.02) | −0.15*** (0.02) | −0.05* (0.02) |
| Log (physicians per 1000) | | 1.27* (0.61) | | 1.95** (0.66) |
| Mean years of schooling | | −0.41 (0.21) | | −0.69* (0.27) |
| Log (GDP per capita) | | 1.82*** (0.46) | | 3.29*** (0.58) |
| GSNI data collection period 2010–2014 | | 1.30 (0.91) | | 0.95 (1.14) |
| MMR | | −0.01*** (0.003) | | |
| $R^2$ | 0.43 | 0.86 | 0.39 | 0.76 |
| Adjusted $R^2$ | 0.42 | 0.84 | 0.38 | 0.74 |
| F statistic | 55.39*** (df=1; 73) | 60.02*** (df=6; 59) | 46.59*** (df=1; 73) | 39.35*** (df=5; 61) |

Numbers in brackets are SEs.
*P values<0.05, **p<0.01,***p<0.001.
CVD, cardiovascular disease; GDP, gross domestic product; GSNI2, Gender Social Norms Index 2; MMR, maternal mortality ratio.

results of sensitivity analysis demonstrated no change in interpretation of results with use of GSNI1 variable as an alternative to GSNI2. The results of this analysis are available on request.

## DISCUSSION

We found a statistically significant relationship between higher country levels of biased gender social norms as measured by the GSNI and higher CVD mortality rates and lower life expectancy for both women and men. These results remained statistically significant after adjusting for country-level economic, education and healthcare status measures. There was no statistically significant relationship demonstrated between the GSNI and female to male CVD mortality ratios within the adjusted model.

Despite a considerable body of research on the relationship between gender inequalities and health, there has been little to no previous quantitative investigation into how gender social norms may link to CVD and life expectancy. Kim et al have linked greater gender inequality as measured by the Gender Inequality Index to higher female to male stroke mortality ratios and lower life expectancy, with use of similar ecological study designs.[37–40] However, some of these analyses did not control for confounding factors leaving results open to significant bias. The choice of covariates used in our study were designed to control for likely confounders within the realms of economics, education and healthcare provision at the country level, all known to have relationships with population health.[28–30]

Additionally, previous research in the area of gender inequality and health has made use of the Gender Inequality Index as the primary explanatory variable.[37–40] However, linking this particular index to health outcomes is problematic as the Gender Inequality Index includes maternal mortality ratio, which is inherently related to health outcomes. The use of the GSNI within our study avoids this limitation. Furthermore, the Gender Inequality Index is a measure of the outcomes of gender inequality, for example, adolescent birth rates and female shares of parliamentary seats, rather than a measure of the social norms held by populations which influence these unequal outcomes, this study aimed to explore the latter.

O'Neil et al have proposed pathways through which gender influences norms, roles, attitudes and knowledge to affect exposure to psychosocial stressors and health behaviours which influence risk of CVD. For example, there is evidence that boys are socialised to take up more physical activity compared with girls, and adolescent girls are more likely to use smoking as a weight loss tool due to social pressures.[19] Additionally, the authors argue that traditional gender roles at home and work may be associated with worse cardiovascular health, including the expectations for men to be the sole breadwinner and women to take on caregiving responsibilities.[19] The psychological impact of sexual abuse and gender discrimination has also been identified as a likely risk factor for CVD.[19] The results of our study support this theoretical framework with the suggestion at a country level of a significant association between gender social norms and population-level CVD outcomes and life expectancy.

### Strengths and limitations

The strengths of this study include the robustness of the underlying data used, its global coverage and novel use of the GSNI. Additionally, important potentially confounding factors have been adjusted for within the multivariable models. Sensitivity analysis was conducted

demonstrating that all results were consistent across differing data time points apart from the male life expectancy outcome.

There are, however, important limitations to consider when interpreting the results of this analysis. Gender social norms are a complex social phenomenon and therefore challenging to measure and quantify, and the underlying data used to calculate the GSNI will likely be affected by selection bias, non-response bias and social desirability bias. Additionally, GSNI values were not available for all countries globally.

The GSNI results included in this study were collected over a long period of time (2005–2014) and therefore there may be lack of standardisation between the data collected at different time points. It is hoped that including the GSNI collection period within the multivariable model will adjust for any systematic differences. However, while the GSNI values were not found to be statistically significantly different over time between the two waves of data collection, only 31 countries had data available for both time points. It may be the case that a country which had GSNI data collected in 2005 may have significantly changed norms in 2014, which is the time point used for the other data points within the main analysis. The sensitivity analysis ran the model with data from 2017, the results of which did not change in direction or significance apart from the male life expectancy outcome. However, it is noted that if GSNI data was collected in 2005 that remains over a decade in which gender social norms may have changed.

Ecological studies are used to measure population-level variables and to influence policy at a regional and national scale.[41] Recognising gender social norms as pervasive in the environment and influential beyond the individual, an ecological study was deemed an appropriate design for this research question. However, it is well recognised that associations at the ecological level may not be seen at an individual level.[42] It could be argued that understanding the relationship between individually held gender social norms and individual health is an important but different question. However, the results of this research must be interpreted with caution in this regard.

Finally, the variables used as covariates with an unclear causal pathway may have introduced bias into the models. It is noted that the models demonstrate potentially unexpected covariate results including higher mean years of schooling and higher numbers of physicians per population associated with higher CVD mortality. This may be the result of countries with improving levels of such determinants experiencing increased burden from non-communicable diseases like CVD due to increased survival from other causes. As with any ecological study, it is essential to recognise this design cannot lead to any firm conclusions about a causal relationship between variables. However, our results suggest a strong positive and statistically significant association between gender social norms and CVD mortality that merits further exploration.

## Implications

The results of this research imply there is an association between country level gender social norms and CVD mortality and life expectancy for men and women. The lack of association between gender social norms and female to male CVD mortality ratio suggests that gender social norms may affect both male and female CVD mortality rates equally. Based on the modelling results, a 10% improvement in a country's GSNI score could be associated with a 50 per 100 000 reduction in CVD mortality rates for both men and women. Additionally, the total variation observed between countries in GSNI value could account for 6.3 years of life expectancy difference in women and 4.5 years in men. This supports the hypothesis that gender social norms are a social determinant of CVD and overall health more broadly. It is likely that there are additional pathways at play affecting the relationship between gender social norms and life expectancy besides its association with CVD mortality and this warrants further investigation.

The relationships between gender social norms and CVD outcomes and life expectancy should be explored further with use of alternative measures of gender social norms, analysis over time and explore options for mediation analysis on the individual-level. Country case studies could also be explored for those with extreme values of the GSNI. The findings of this research suggest that it would be justified to investigate biased gender social norms as a phenomenon that may impact negatively on the entire population's health, in wide ranging ways, particularly in the area of non-communicable diseases. Therefore, this research should be added to the ever growing body of evidence that improving biased gender social norms and gender equality can contribute to better population health and should be a priority for policy decision-makers as set out within the Sustainable Development Goals.

**Contributors** IL, AH and MO conceived and designed the study. IL acquired and managed the data and completed all statistical analysis, which was reviewed by AH and MO. IL wrote the first draft and is the guarantor for this study. SSK and ML contributed to the critical discussion of the results and manuscript drafting. The corresponding author attests that all listed authors meet authorship criteria and that no others meeting the criteria have been omitted.

**Funding** IL is an Academic Clinical Fellow funded by the UK National Institute for Health and Care Research in conjunction with Lancaster University. They are employed by St Helens and Knowsley NHS Foundation Trust. AH is funded by a departmental studentship at the University of Liverpool and supported by the UK National Institute for Health and Care Research School for Public Health (Grant Reference Number PD-SPH-2015-10025). SSK receives funding from the National Heart, Lung and Blood Institute and the American Heart Association. These funders had no role in the study design, analysis or presentation of results. The views expressed are those of the authors and not necessarily those of the National Institute for Health and Care Research, the NHS, the National Heart, Lung and Blood Institute or the American Heart Association. MO is partly funded (among other grants) by ESRC grant ES/W007932/1: Developing public health policies for the out of home food sector to improve diet and reduce obesity, and NIHR grant NIHR130258: Unmet need for healthcare.

**Competing interests** None declared.

**Patient and public involvement** Patients and/or the public were not involved in the design, or conduct, or reporting, or dissemination plans of this research.

**Patient consent for publication** Not applicable.

**Ethics approval** All data used in this research was non-identifiable and freely available. Therefore, after submission to the University of Liverpool ethics review board (reference number 9811) the research was approved without the need for review by the ethics committee.

**Provenance and peer review** Not commissioned; externally peer reviewed.

**Data availability statement** Data are available in a public, open access repository. The data used in this analysis are openly available from the Institute for Health Metrics and Evaluation, The World Bank, The Global Health Observatory, the United Nations Development Programme and Our World in Data. Code used for analysis is available via GitHub at https://github.com/ilsumme/gsn-cvd-le.

**ORCID iD**
Iona Lyell http://orcid.org/0000-0002-5923-2231

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
