## [Reviewer comments · BMJ Open]

ARTICLE DETAILS

TITLE (PROVISIONAL)	The association between gender social norms and cardiovascular disease mortality and life expectancy: an ecological study
AUTHORS	Lyell, Iona; Khan, S; Limmer, Mark; O'Flaherty, M; Head, Anna

VERSION 1 – REVIEW

REVIEWER	Mohus, Randi Marie Norwegian University of Science and Technology, Department for Circulation and Medical Imaging
REVIEW RETURNED	29-Jun-2022

GENERAL COMMENTS	Review comments: The manuscript presents a study of the associations between gender social norms and cardiovascular disease (CVD) mortality rates in men and women and the total age-standardised mortality rate, the female to male CVD mortality ratios, and life expectancy in men and women. The study is based on a global data from many countries with data on gender social norms as exposure variables, and outcome data on CVD mortality from the Global Burden of Disease study. The overall impression of this manuscript is that it covers an important topic which has not been thoroughly elucidated previously. The study used an ecological study design and included potential confounding factors in their multivariate analyses which adds to the strengths of this study. I find the study to be well suited to address the study questions. The most important findings in this study are that CVD related mortality is higher in both men and women in countries with higher gender biased social norms, and also the finding that both men and women have lower life expectancy in these countries. I add to my commentary that my area of expertise is sex differences in infectious diseases and that the ecological study framework is not my field in epidemiology. The Editor should include other reviewers with more knowledge in ecological studies to ensure proper evaluation of the statistical analyses performed. I have some comments: The exposure variable which is the GSNI is a novel indicator of gender social norms and has not been widely used before. I miss some argumentation of how the authors decided to use the cut-off of GSNI2 as the primary explanatory variable. I also find the inclusion of covariates such as GDP/capita, Physicians/1000 population and mean years of schooling very informative. The authors have mentioned some important limitations of their study and maybe the lack of repeated measurements of the GSNI in the later period reflect the most important limitation of this study. Nevertheless, I find the results from GSNI collected from 2005-2014
---

	very informative, and based on the presented results, new efforts to collect global numbers of GSNI are of utmost importance. Minor: For the results I miss some information on the particular countries included, for example in Figure 1 and 2 – the scatter plots could have included all country names (not only the outliers). At least should this be included as supplemental figures or as a table with individual country results. We only get to know the “outliers”, whereas we do not have any comparison on how the other 75 countries comply. I actually would recommend providing this information as this would be very informative to health authorities and public health leaders also in the countries rated middle or best, and it would be very informative for future studies. Minor: In all tables except Table 1, I would recommend adding an explanation of GDP to the abbreviation list. In summary I find the study to be well conducted and important for future in-depth studies of gender social norms and their influence on population health outcomes.
--	---

REVIEWER	Lin, Ro-Ting Harvard T.H. Chan School of Public Health, Department of Global Health and Population
REVIEW RETURNED	18-Aug-2022

GENERAL COMMENTS	Thank you very much for inviting me to review this manuscript. This ecological study analyzed the association between gender social norms and cardiovascular disease mortality and life expectancy in 75 countries. The major question is the unclear time interval between independent and dependent variables. Several questions and comments should be clarified and addressed before further considering this manuscript for publication.  1. Page 6, lines 24–32: “Whilst there are theoretical explanations for a connection ... at a population level, this has not yet been explored ... use of country level data.” What is the theoretical explanation for results using population-level data? Should the theoretical explanation be the same or different when using country-level data? The rationale for conducting a country-level investigation should be elaborated. 2. Page 6, lines 46–56: “Examples of connections between gender social norms and adverse health outcomes...” Examples have been provided. However, the hypothetical pathways for how gender social norms inequality affects cardiovascular disease mortality and life expectancy is unclear. The authors addressed some possibilities in the discussion section (i.e., Implication section). I'd like to suggest a brief explanation of the rationale in the introductory section. 3. Page 8, paragraph 2: The GSNI dataset includes the share of people with at least one bias (GSNI) and the share of people with at least two biases (GSNI2). Why choose GSNI2 over GSNI? Are there any differences in the results when using two different metrics? The robustness can be verified using sensitivity analysis. 4. Following the above comments, it is unclear why the Gender Inequality Index was not chosen. Although the authors mentioned that the Gender Inequality Index covered maternal health, one of the strengths of the Gender Inequality Index is that it contained annual information, whereas the GNSI did not. As mentioned earlier, the main problem of this study is the unclear time interval between the independent and dependent variables. Has the time interval been properly considered in the analysis?
---

	5. The year for each variable should be clarified and specified (in text, tables, and figures). For example, the life expectancy mentioned in line 29 on page 11, should it be the life expectancy in 2017 or the life expectancy in 2019? In addition, Table 1 seems to show only the publication year of the data source, not the year each indicator represents. 6. Page 14, line 29: Qatar was treated as an outlier. What is the definition of an outlier in this study? 7. Tables 2 and 3: The meaning of “GSNI data collection period 2010-2014” is unclear. Does it represent time for GSNI data? Or, does it represent the GSNI score? Also, why do parameter estimates show negative values? 8. The “Implication” section is too long. In line with the above comment, some of the content can be briefly explained in the Introduction section.
--	--

REVIEWER	Musa, Ahmad Monash University Malaysia, Clinical Sciences
REVIEW RETURNED	7-Oct-2022

GENERAL COMMENTS	Very well-researched and well-written manuscript.
---

VERSION 1 – AUTHOR RESPONSE

Reviewer: 1

Dr. Randi Marie Mohus, Norwegian University of Science and Technology Comments to the Author:
BMJ-open

The association between gender social norms and cardiovascular disease mortality and life expectancy: an ecological study

Manuscript ID: BMJopen-2022-065486

Trondheim, 26 June 2022 Thank you for the opportunity to review this article.

Review comments:

The manuscript presents a study of the associations between gender social norms and cardiovascular disease (CVD) mortality rates in men and women and the total age-standardised mortality rate, the female to male CVD mortality ratios, and life expectancy in men and women. The study is based on a global data from many countries with data on gender social norms as exposure variables, and outcome data on CVD mortality from the Global Burden of Disease study. The overall impression of this manuscript is that it covers an important topic which has not been thoroughly elucidated previously.

The study used an ecological study design and included potential confounding factors in their multivariate analyses which adds to the strengths of this study. I find the study to be well suited to address the study questions. The most important findings in this study are that CVD related mortality is higher in both men and women in countries with higher gender biased social norms, and also the finding that both men and women have lower life expectancy in these countries.

I add to my commentary that my area of expertise is sex differences in infectious diseases and that the ecological study framework is not my field in epidemiology. The Editor should include other reviewers with more knowledge in ecological studies to ensure proper evaluation of the statistical analyses performed.

I have some comments:

The exposure variable which is the GSNI is a novel indicator of gender social norms and has not been widely used before. I miss some argumentation of how the authors decided to use the cut-off of

GSNI2 as the primary explanatory variable.

*A statement has been added to the methods section regarding this: “The share of people with at least two gender biases (GSNI2) was chosen as the primary explanatory variable as it was presumed this would demonstrate a stronger relationship if there was one to be found compared with the share of people with one bias.”

I also find the inclusion of covariates such as GDP/capita, Physicians/1000 population and mean years of schooling very informative.

The authors have mentioned some important limitations of their study and maybe the lack of repeated measurements of the GSNI in the later period reflect the most important limitation of this study. Nevertheless, I find the results from GSNI collected from 2005-2014 very informative, and based on the presented results, new efforts to collect global numbers of GSNI are of utmost importance.

Minor:

For the results I miss some information on the particular countries included, for example in Figure 1 and 2 – the scatter plots could have included all country names (not only the outliers). At least should this be included as supplemental figures or as a table with individual country results. We only get to know the “outliers”, whereas we do not have any comparison on how the other 75 countries comply. I actually would recommend providing this information as this would be very informative to health authorities and public health leaders also in the countries rated middle or best, and it would be very informative for future studies.

*Additional scatter plot graphs have been added to the supplemental material which include points labelled by country. An interactive graph has also been made available via GitHub (linked to via the supplemental material) where the points can be hovered over to see which point corresponds to which country <https://ilsumme.github.io/gsn-cvd-le/> . It has also been made clear which table in the supplemental material corresponds to the list of countries included in the scatter plots. It was decided not to include the labelled graphs in the primary figures as it is more difficult to see the underlying pattern of association which was the primary objective of the figures. The few countries labelled are the points with extreme values as it was thought this would be useful additional information for readers.

Minor:

In all tables except Table 1, I would recommend adding an explanation of GDP to the abbreviation list.

*An expansion of this abbreviation has been added to table legends

In summary I find the study to be well conducted and important for future in-depth studies of gender social norms and their influence on population health outcomes.

Yours sincerely,
Randi Marie Mohus MD
Senior consultant
Clinic of Anesthesia and Intensive Care
St. Olavs hospital,
Trondheim,
Norway

Reviewer: 2

Dr. Ro-Ting Lin, Harvard T.H. Chan School of Public Health Comments to the Author:

Thank you very much for inviting me to review this manuscript. This ecological study analyzed the

association between gender social norms and cardiovascular disease mortality and life expectancy in 75 countries. The major question is the unclear time interval between independent and dependent variables. Several questions and comments should be clarified and addressed before further considering this manuscript for publication.

1. Page 6, lines 24–32: “Whilst there are theoretical explanations for a connection ... at a population level, this has not yet been explored ... use of country level data.” What is the theoretical explanation for results using population-level data? Should the theoretical explanation be the same or different when using country-level data? The rationale for conducting a country-level investigation should be elaborated.

*The introduction has now been re-organised and theory section strengthened to make clear the theory context before introducing the research question. Please see a section of this below: “Theoretical explanations exist to link gender and gender social norms to CVD outcomes including in healthcare, health behaviours and wider determinants of health¹⁹. These include a recognition of gender bias in research and treatment of CVD, an understanding that gender social norms influence patterns of CVD risk factors e.g. stress and smoking, and their impact on important social determinants of CVD including education and employment^{17, 20, 21}. Therefore, whilst there are theoretical explanations for a connection between gender social norms and CVD mortality at a population level¹⁹, this has not yet been explored in an ecological framework with use of country level data.

It is theorised that increased levels of biased gender social norms may be associated with determinants and behaviours harmful to cardiovascular health on a population-level, and therefore an increased CVD burden and reduced life expectancy more broadly for both women and men^{19, 22}. Therefore, this study aimed to explore the association between gender social norms, as measured by the Gender Social Norms Index and CVD mortality and life expectancy at the country level.”

2. Page 6, lines 46–56: “Examples of connections between gender social norms and adverse health outcomes...” Examples have been provided. However, the hypothetical pathways for how gender social norms inequality affects cardiovascular disease mortality and life expectancy is unclear. The authors addressed some possibilities in the discussion section (i.e., Implication section). I'd like to suggest a brief explanation of the rationale in the introductory section.

*Please see above response to comment 1.

3. Page 8, paragraph 2: The GSNI dataset includes the share of people with at least one bias (GSNI) and the share of people with at least two biases (GSNI2). Why choose GSNI2 over GSNI? Are there any differences in the results when using two different metrics? The robustness can be verified using sensitivity analysis.

*A statement has been added to the methods section regarding the choice of GSNI2 over GSNI1: “The share of people with at least two gender biases (GSNI2) was chosen as the primary explanatory variable as it was presumed this would demonstrate a stronger relationship if there was one to be found compared with the share of people with one bias.”

The methods now include the addition of sensitivity analysis with GSNI1 and results added as per below:

“GSNI1 and GSNI2 were strongly positively correlated ($\rho = 0.98$, $p\text{-value} = <0.05$) and the results of sensitivity analysis demonstrated no change in interpretation of results with use of GSNI1 variable as an alternative to GSNI2. The results of this analysis are available on request.”

4. Following the above comments, it is unclear why the Gender Inequality Index was not chosen.

Although the authors mentioned that the Gender Inequality Index covered maternal health, one of the strengths of the Gender Inequality Index is that it contained annual information, whereas the GSNI did not. As mentioned earlier, the main problem of this study is the unclear time interval between the independent and dependent variables. Has the time interval been properly considered in the analysis?

*The reason why the Gender Inequality Index was not chosen has been expanded on in the discussion as per below. The choice of timepoint of variables has been justified in the methods section based on GSNI collection period, and sensitivity analysis was undertaken with the most recent time points available to explore potential impact of time interval chosen. The limitation of not having additional time points available for GSNI has been noted in the key strengths and limitations. "Additionally, previous research in the area of gender inequality and health has made use of the Gender Inequality Index as the primary explanatory variable³⁷⁻⁴⁰. However, linking this particular index to health outcomes is problematic as the Gender Inequality Index includes maternal mortality ratio, which is inherently related to health outcomes. The use of the Gender Social Norms Index within our study avoids this limitation. Furthermore, the Gender Inequality Index is a measure of the outcomes of gender inequality e.g. adolescent birth rates and female shares of parliamentary seats, rather than a measure of the social norms held by populations which influence these unequal outcomes, this study aimed to explore the latter."

5. The year for each variable should be clarified and specified (in text, tables, and figures). For example, the life expectancy mentioned in line 29 on page 11, should it be the life expectancy in 2017 or the life expectancy in 2019? In addition, Table 1 seems to show only the publication year of the data source, not the year each indicator represents.

*The year that each variable is collected for has been added to table 1 to make it clearer which time point is used for the main analysis. The year has also been added for the outcomes in all results tables. It is also laid out in the methods in regard to what time points are used in the main analysis and in the sensitivity analysis.

6. Page 14, line 29: Qatar was treated as an outlier. What is the definition of an outlier in this study?

*The following had been added to the methods:

"Residuals vs leverage plots and Cook's distance were used to identify potential outliers or influential observations."

7. Tables 2 and 3: The meaning of "GSNI data collection period 2010-2014" is unclear. Does it represent time for GSNI data? Or, does it represent the GSNI score? Also, why do parameter estimates show negative values?

*"GSNI data collection period 2010-2014" refers to the timeframe over which the data was collected for the GSNI which is laid out within the methods section please see below. The negative coefficient for the GSNI period 2010-2014 is likely due to the different mix of countries that had data collected over the two different time points. Whilst it was important to take the two different time points of GSNI collection into consideration in the analysis, it is not useful to look at the relationship between GSNI collection period and the outcome variables in isolation as there is a variable mix of countries included in these two collection time points. This is also expanded on in the discussion please see below.

Methods:

(Primary explanatory variable section which explains the two collection periods of GSNI) "The [GSNI] data was collected in two waves of the survey undertaken between 2005-2009 and 2010-2014."

(Covariates section which justifies its inclusion in the model) "It is possible that the wave of GSNI data

collection may have influenced the results e.g. through passage of time or changes in methodology. Therefore, this variable was also included within the model.”

Discussion:

“The GSNI results included in this study were collected over a long period of time (2005-2014) and therefore there may be lack of standardisation between the data collected at different time points. It is hoped that including the GSNI collection period within the multivariable model will adjust for any systematic differences. However, whilst the GSNI values were not found to be statistically significantly different over time between the 2 waves of data collection, only 31 countries had data available for both time points. It may be the case that a country which had GSNI data collected in 2005 may have significantly changed norms in 2014, which is the time point used for the other data points within the main analysis.”

8. The “Implication” section is too long. In line with the above comment, some of the content can be briefly explained in the Introduction section.

*This has been actioned please see above and the implications section edited down substantially.

Reviewer: 3

Dr. Ahmad Musa, Monash University Malaysia Comments to the Author:

Very well-researched and well-written manuscript.

VERSION 2 – REVIEW

REVIEWER	Mohus, Randi Marie Norwegian University of Science and Technology, Department for Circulation and Medical Imaging
REVIEW RETURNED	26-Mar-2023
GENERAL COMMENTS	No special comments to the author.
REVIEWER	Lin, Ro-Ting Harvard T.H. Chan School of Public Health, Department of Global Health and Population
REVIEW RETURNED	25-Mar-2023
GENERAL COMMENTS	Thank you very much for revising the manuscript. After reading the revision, I believe that changes you have made have greatly improved the clarify, robustness, and presentation of your research. Therefore, I'd like to suggest that the manuscript is now suitable for publication.